# Effects of arsenic on the topology and solubility of promyelocytic leukemia (PML)-nuclear bodies

**Seishiro Hirano** [1]*, **Osamu Udagawa**[1]

Center for Health and Environmental Risk Research, National Institute for Environmental Studies, Tsukuba, Ibaraki, Japan

* seishiro@nies.go.jp

## Abstract

Promyelocytic leukemia (PML) proteins are involved in the pathogenesis of acute promyelocytic leukemia (APL). Trivalent arsenic ($As^{3+}$) is known to cure APL by binding to cysteine residues of PML and enhance the degradation of PML-retinoic acid receptor α (RARα), a t (15;17) gene translocation product in APL cells, and restore PML-nuclear bodies (NBs). The size, number, and shape of PML-NBs vary among cell types and during cell division. However, topological changes of PML-NBs in $As^{3+}$-exposed cells have not been well-documented. We report that $As^{3+}$-induced solubility shift underlies rapid SUMOylation of PML and late agglomeration of PML-NBs. Most PML-NBs were toroidal and granular dot-like in *GFPPML*-transduced CHO-K1 and HEK293 cells, respectively. Exposure to $As^{3+}$ and antimony ($Sb^{3+}$) greatly reduced the solubility of PML and enhanced SUMOylation within 2 h in the absence of changes in the number and size of PML-NBs. However, the prolonged exposure to $As^{3+}$ and $Sb^{3+}$ resulted in agglomeration of PML-NBs. Exposure to bismuth ($Bi^{3+}$), another Group 15 element, did not induce any of these changes. ML792, a SUMO activation inhibitor, reduced the number of PML-NBs and increased the size of the NBs, but had little effect on the $As^{3+}$-induced solubility change of PML. These results warrant the importance of $As^{3+}$- or $Sb^{3+}$-induced solubility shift of PML for the regulation intranuclear dynamics of PML-NBs.

## Introduction

Promyelocytic leukemia proteins (PMLs) are proapoptotic molecules involved in cell proliferation, senescence, and tumor suppression [1, 2]. The human PML family comprises six nuclear isotypes (I-VI) and one cytosolic isotype (VII) [3–5]. The residence-time of PML in PML nuclear bodies (PML-NBs) ranges from several minutes to one hour, whereas that of other PML-NB client proteins such as Sp100 and Daxx varies from several seconds to one minute [6], which led to the notion that PML is a scaffold protein for PML-NBs. Sp100 is not essential for the maintenance of PML-NBs in human embryonic NT2 cells and the C-terminal region of Daxx is sufficient for interaction with PML [7]. The mutation of three small ubiquitin-like modifier (SUMO) conjugation sites of PML (K65, K160, and K490) did not affect the

**Data Availability Statement:** All relevant data are within the manuscript and its Supporting Information files.

**Funding:** This work was partially supported by a Grant-in-Aid from the Japan Society for the

Promotion of Science (16K15386 given to SH). The funders had no role in study design, data collection and analysis, decision to publish, or preparation of the manuscript.

**Competing interests:** The authors have declared that no competing interests exist.

formation of PML-NBs, although this mutant could recruit neither Sp100 nor Daxx [8]. The SUMOylation of PML is thought to stabilize and promote the assembly of PML-NBs, since free PML tends to disperse throughout the nucleoplasm and SUMO1-conjugated PML localizes to PML-NBs [9]. PML-NBs are structurally and positionally stable in interphase and appear to associate with euchromatin via SUMO molecules [10].

PML is a member of RBCC/tripartite motif (TRIM) family. Cysteines in both B-box zinc finger domains ($B_1$ and $B_2$) are required for PML-NB formation [11]. Trivalent arsenic ($As^{3+}$) binds to cysteine residues of the RING and $B_2$ domains [12] and the core region of RING is requisite for the solubility change and SUMOylation of PML in response to exposure to $As^{3+}$ [13, 14]. $C_{212}$ in the $B_2$ domain is required for $As^{3+}$-induced degradation of PML-retinoic acid receptor α (PML-RARα), a t(15;17) gene translocation product [15]. The subdomain comprising $F_{52}Q_{53}F_{54}$, and $L_{73}$ in the RING motif (aa 55–99) are unique to PML and are absent in other TRIM family members. The unique subdomain appears to play a role in PML tetramerization and the subsequent PML-NB formation, and also in $As^{3+}$-induced SUMOylation of PML [16]. Together, intact zinc finger domains of RBCC and the unique RING motif of PML are all necessary for functional PML-NBs and biochemical responses of PML to $As^{3+}$. PML-RARα functions as a dominant negative form and causes acute promyelocytic leukemia (APL) [17, 18]. Both $As^{3+}$ and all-trans retinoic acid (ATRA) have been used to treat APL [19, 20]. $As^{3+}$-binding to PML renders PML-RARα susceptible to ubiquitin-mediated degradation by RNF4 which non-covalently binds to polySUMOylated PML via SUMO-interacting motifs (SIMs) [4, 21].

Canonical PML-NBs are present in the nucleus as dense granular bodies or hollow toroid-like oblate (doughnut shaped) spheres [22], and the shape and size of PML-NBs vary depending on the cell type [23]. Non-spherical forms of PML-NBs have also been reported. Fiber-like PML-NBs are formed along with doughnut-like PML-NBs in the nuclei of IDH4 cells, human fibroblasts immortalized with the dexamethasone-inducible SV40 [24]. Nuclear envelope-associated linear and rod-like PML bodies reportedly arise transiently during the very early transitions of human ES cells towards cell-type commitment [25]. Progerin, a truncated form of lamin A, has been shown to colocalize with aberrant toroid-like and thread-like PML-NBs in fibroblasts obtained from Hutchinson-Gilford progeria syndrome patients [26], suggesting that the topology of PML-NBs may be linked with some pathological consequences.

Although PML-NBs are stable in interphase nuclei, they aggregate to form mitotic accumulation of PML proteins (MAPPs) after nuclear membrane breakdown. MAPPs associate with nuclear pore components in daughter cells after mitosis. At this stage the aggregated PML bodies are called cytoplasmic assemblies of PML and nucleoporins (CyPNs) and reside in perinuclear regions [27, 28]. MAPPs are inherited asymmetrically in daughter HaCaT cells after mitosis [29], and daughter cells that receive the majority of PML-NBs exhibit increased stemness in primary human keratinocytes [28]. Since PML-NBs feature phase-separated protein condensates in the nucleus [30, 31], it is of interest to investigate how their biochemical/biophysical properties change upon cell division and exposure to $As^{3+}$.

In the present work, we performed a topological characterization of PML-NBs using CHO-K1 and HEK293 cells stably transduced with GFP-conjugated *PML-VI*. PML-VI is the only nuclear PML isoform that lacks a SIM motif, and therefore the SUMO-SIM interaction between PML and the other proteins, which is implicated in liquid-liquid phase separation (LLPS) [32], occurs only when PML-VI is SUMOylated. PML-NBs were toroidal in CHO cells and granular dot-like in HEK cells. Two-hour exposure of these cells to $As^{3+}$ induced a robust SUMOylation of PML-VI although the size and number of PML-NBs were not changed. The solubility shift of PML-VI occurred prior to SUMOylation of this protein. We also show that inhibition of SUMOylation rather than the $As^{3+}$-induced robust SUMOylation regulated the number and size of PML-NBs.

## Materials and methods

### Chemicals

TAK243 (Ub-activating E1 enzyme inhibitor, Selleckchem, Houston, TX) and ML792 (SUMO E1 inhibitor, MedKoo Bioscience, Morrisville, NC) were dissolved in DMSO and used at a final concentration of 0.1% DMSO. A 20S proteasome assay kit was purchased from (Enzo, Famingdale, NY). Bismuth (III) nitrate pentahydrate and antimony (III) chloride were purchased from Wako (Osaka, Japan). They were dissolved in 0.1 M citrate buffer (pH 7.4) at 10 mM and used as stock solutions. Sodium *m*-arsenite was purchased from Sigma-Aldrich (St. Louis, MO) and dissolved in PBS solution. These solutions were sterilized using a syringe filter (0.45 μm pore size). Unless otherwise specified, common chemicals of analytical grade including cadmium sulfate were obtained from Sigma-Aldrich or Wako.

### Cells and GFP monitoring

The human *PML-VI* gene engineered into the pcDNA6.2N-EmGFP DEST vector was transduced into HEK293 and CHO-K1 cells [33]. Stable transfectants were selected using blasticidine S and designated as HEKGFPPML and CHOGFPPML cells, respectively. The cells were cultured in glass-bottom culture dishes and GFP fluorescence was monitored by confocal laser scanning microscopy (TCS-SP5, Leica Microsystems, Solms, Germany) or fluorescence microscopy (Eclipse TS100, Nikon, Kanagawa, Japan).

### Protein extraction and western blotting

The cell monolayers were rinsed with Hank's balanced salt solution (HBSS) and lysed with ice-cold RIPA buffer (Santa Cruz, Dallas, TX) containing protease (Santa Cruz) and phosphatase inhibitor cocktails (Calbiochem/Merk Millipore, San Diego, CA) on ice for 10 min. The lysate was centrifuged at 9,000 *g* for 5 min at 4˚C. The supernatant was collected and labelled as the RIPA-soluble fraction (Sol). The pellet, which contained nucleic acids, was rinsed with PBS and treated with Benzonase® nuclease (250 U/mL in Tris buffer, Santa Cruz) of the same volume as the RIPA buffer at 25˚C for 2 h with intermittent mixing (Thermomixer comfort, Eppendorf, Wesseling-Berzdorf, Germany). The digested sample was labelled as the RIPA-insoluble fraction (Ins). The following antibodies were used for immunoblot analyses, with dilution rates of 1:750 for primary antibodies and 1:2500 for secondary antibodies in Can-Get-Signal™ solution (TOYOBO, Osaka, Japan): anti-PML antibody (Bethyl, A301-167A, rabbit polyclonal); anti-SUMO2/3 (MBL, 1E7, mouse, monoclonal); anti-multi ubiquitin (MBL, FK2, mouse, monoclonal); anti-SUMO1 (CST, C9H1, rabbit, monoclonal); anti-UBA2 (CST, D15C11, rabbit, monoclonal); anti-GFP (Abcam, goat, polyclonal); HRP-conjugated anti-histone H3 (CST, #12648, rabbit, monoclonal); HRP-conjugated anti-α-tubulin (MBL, #PM054-7, rabbit, polyclonal); HRP-conjugated goat anti-rabbit IgG (Santa Cruz, sc-2054); HRP-conjugated goat anti-mouse IgG (Santa Cruz, sc-2055); HRP-conjugated mouse anti-goat IgG (Santa Cruz, sc-2354). Aliquots of the RIPA-soluble fraction and the nuclease-treated pellet suspension (RIPA-insoluble fraction) were mixed with LDS sample buffer (1x TBS, 10% glycerol, 0.015% EDTA, 50 mM DTT, and 2% LDS) and heated 95˚C for 5 min. Proteins in the samples were resolved by LDS (SDS)-PAGE (4–12%) and electroblotted onto PVDF membranes. The membranes were blocked with PVDF Blocking Reagent (TOYOBO) before probing with antibodies. Signals on the ECL (Prime, GE Healthcare, Buckinghamshire, UK)-treated PVDF membrane were then captured with CCD cameras (Lumino Imaging Analyzer, FAS-1100, TOYOBO; Amersham ImageQuant 800, GE Healthcare, Uppsala, Sweden).

## Immunofluorescent staining

CHOGFPPML cells were grown in an 8-well chamber slide (Millicell EX slide, Merck Milli-pore, Burlington, MA) to early confluency. The cells were washed with warmed (37˚C) HBSS and fixed with 3.7% formaldehyde solution for 10 min, permeabilized with 0.1% Triton X-100 for 10 min, and treated with Image-iT™ FX Signal Enhancer (Invitrogen-ThremoFischer, Carlsbad, CA) for another 10 min. The cells were immunostained with Alexa Fluor® 594-labeled anti-SUMO-2/3 for 45 min. Anti-SUMO2/3 antibody (MBL) was labelled with Alexa Fluor® 594 using an Alexa Fluor™ 594 Antibody Labeling Kit (Invitrogen-Thermo-Fischer) and diluted to 1/200 in Can-Get-Signal™ Immunostain solutions (TOYOBO). DAPI was used to counter-stain the nuclei. The fluorescent images were captured using a fluores-cence microscope (Eclipse 80i, Nikon, Tokyo) and the digital images were assembled using Adobe PHOTOSHOP® software.

## Immunoprecipitation

The cells were lysed with cold RIPA solution and clear supernatants were obtained by centrifu-gation (9000 $g$, 5 min). The protein concentration of the supernatant was adjusted to 2 mg/mL with RIPA. The sample was diluted to one fourth the initial concentration with 150 mM Tris-HCl solution (pH 7.2) and reacted with anti-GFP mAb-magnetic beads (MBL) or control IgG2a-magenitic beads (MBL) at 4–8˚C for 2 h with intermittent vortexing. The magnetic beads were washed 3 times with 150 mM Tris-HCl solution (pH 7.2) containing 0.1% NP40 and the pellet was boiled with 2X LDS(SDS)-PAGE sample buffer for 5 min. The sample was centrifuged, then the supernatant was subjected to LDS (SDS)-PAGE and subsequent western blot analysis as described above.

## Cell viability

The cells were pre-cultured to early confluency in a 96-well culture dish, then further cultured in the presence or absence of $As^{3+}$, $Sb^{3+}$, $Bi^{3+}$, or ML792 and/or TAK792. The cells were washed twice with HBSS and the viable cell numbers were assayed colorimetrically by WST-8 (Wako) using a microplate reader (POLARstar OPTIMA, BMG Labtech, Offenburg, Germany).

## Data analyses

Data were presented as means ± SEM. Statistical analyses were performed by ANOVA fol-lowed by Tukey's *post-hoc* test. The number and size analyses of PML-NBs were performed using ImageJ (NIH, https://imagej.nih.gov/ij/) or NIS-Elements software (Nikon). A PML-NB agglomerate was defined as an intranuclear assembly of five or more than five PML-NBs in an aggregated form. A chi-squared ($\chi^2$) test was applied to compare frequencies of cells with PML-NB agglomerates among groups. A probability value of less than 0.05 was accepted as indicative of statistical significance.

# Results

## Topology and dynamics of PML-NBs

Most interphase PML-NBs occurred as toroidal and granular dot-like bodies in CHOGFPPML (Fig 1A) and HEKGFPPML cells (Fig 1B), respectively. The number and size of PML-NBs vary among cells from 4 to 27 in the number and ca. 0.3–1.2 μm in diameter (Fig 1C). Neither the number nor the size was significantly changed by exposure to $As^{3+}$ for 2 h. We next studied time-course changes in the shapes of PML-NBs by confocal laser scanning microscopy using

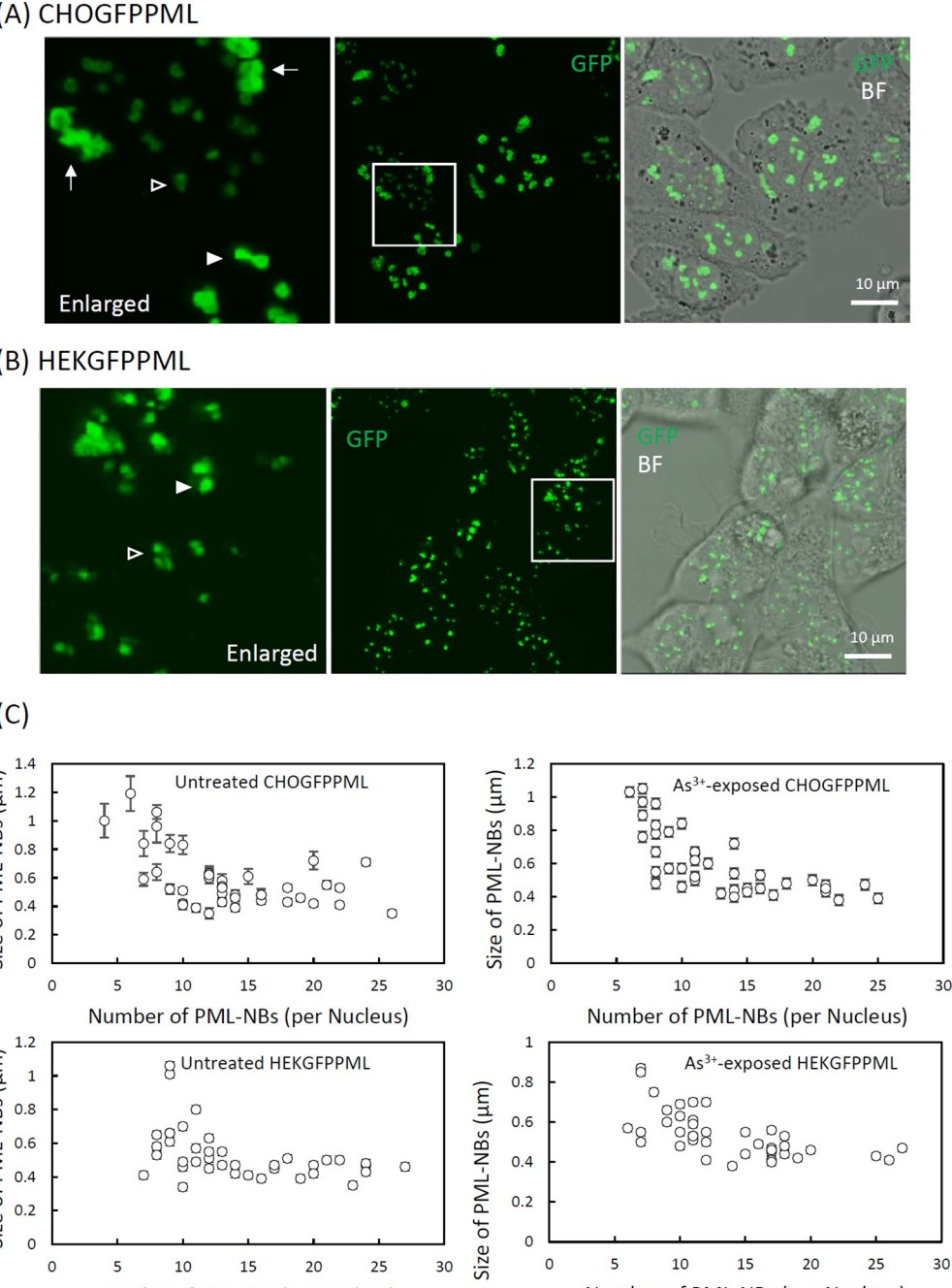

**Fig 1.** The structure of PML–NBs and peri–nuclear PML aggregates in CHOGFPPML (A) and HEKGFPPML cells (B), and the number and size distribution of PML–NBs (C). GFP live images were obtained by confocal laser scanning microscopy in z–stacking mode. The white boxed area was enlarged to visualize the shapes of PML–NBs and peri–nuclear PML aggregates clearly. (A) PML–NBs occurred as either small toroids (open arrowhead) or large toroids (closed arrowhead) in CHOGFPPML cells. The shape of peri–nuclear PML aggregates (arrow) were also toroidal. (B) PML–NBs occurred either as small (open arrowhead) or large granular dots (closed arrowhead) in HEKGFPPML cells. (C) The cells were pre–cultured in an 8–well chamber slide and treated with 3 μM As$^{3+}$ or left untreated for 2 h. The cells were fixed and counterstained with DAPI. The number and size of PML–NBs in each nucleus were measured by fluorescence microscopy. The size of each PML–NB was estimated as the mean of long and short axes. The measurement was performed on 40 nuclei for each group. Each symbol represents the mean ± SEM of PML–NB sizes in each nucleus.

the z-stack mode to further interrogate the dynamic nature of PML-NBs. Fig 2A shows that aggregated PML toroids were found after nuclear membrane breakdown in dividing CHOGFPPML cells (0:00:00). Another large toroidal PML aggregate was freshly formed in the peri-nuclear region via condensation of vague small GFP speckles (arrows) in 87 sec (from 0:11:55 to 0:13:22) as cytokinesis proceeded (See also S1 Fig for a freshly formed peri-nuclear PML toroid). Small nascent PML-NBs (arrowheads) were formed as the daughter cells spread in both untreated (Figs 2A and S1) and As$^{3+}$-exposed CHOGFPPML cells (S2 Fig). Although most PML-NBs of HEKGFPPML cells were granular dot-like, a small percentage (0.5%) of HEKGFPPML cells accommodated unstable toroidal PML-NBs (Fig 2B). The sizes of these toroidal PML-NBs were comparable to those of CHOGFPPML cells. The toroidal PML-NBs in HEKGFPPML cells were, however, gradually dissipated and transformed into small dot-like PML-NBs in 1.5 h (boxed areas), suggesting that the toroidal form of PML-NBs in HEKGFPPML cells are less stable than those in CHOGFPPML cells and eventually transformed into the granular dot-like PML-NBs.

## Effects of As3+, Sb3+, and Bi3+ on PML and intranuclear distribution of PML-NBs

This experiment was performed to study the specificity of As$^{3+}$ as a modifier of intranuclear dynamics of PML-NBs. Both As$^{3+}$ and Sb$^{3+}$ are metalloids, while Bi$^{3+}$ is a heavy metal belonging to Group 15 of the elements. Bi$^{3+}$ was slightly less cytotoxic than either As$^{3+}$ or Sb$^{3+}$ in CHOGFPPML cells (Fig 3A). Exposure to 3 μM As$^{3+}$ or Sb$^{3+}$ for 2 h induced an overt protein solubility change and converted cold RIPA-soluble GFPPML into the RIPA-insoluble form, and SUMOylated GFPPML. However, these biochemical changes of GFPPML were not observed in Bi$^{3+}$-exposed cells (Fig 3B). We next examined a possibility that As$^{3+}$ and Sb$^{3+}$ inhibited the proteasome activity and SUMOylated PML proteins were not eliminated efficiently by ubiquitin-proteasome system (UPS). The 20S proteasome activity was inhibited slightly by 30 μM As$^{3+}$, Sb$^{3+}$, Bi$^{3+}$, and Cd$^{2+}$. However, Bi$^{3+}$ inhibited more efficiently than As$^{3+}$ and Sb$^{3+}$ (S3 Fig), suggesting that the degradation of SUMOylated PML by UPS was not involved in the different SUMOylation level of PML between Bi$^{3+}$-exposed and As$^{3+}$- or Sb$^{3+}$-exposed cells, although we do not deny a possibility that As$^{3+}$ and Sb$^{3+}$ affected the 26S moiety of proteasomes and decreased the UPS activity

Contrary to the explicit biochemical changes in GFPPML, the microscopically observed size, number, and intranuclear distribution of PML-NBs did not remarkably change by the 2-h treatment with either As$^{3+}$ or Sb$^{3+}$. However, after 24-h exposure to 1–3 μM As$^{3+}$ or Sb$^{3+}$, PML-NBs agglomerated in the nucleus of CHOGFPPML cells. Exposure to Bi$^{3+}$ did not cause these microscopic changes (Fig 4). The agglomeration was reduced after removal of As$^{3+}$ or Sb$^{3+}$ from the culture medium (Fig 5A), suggesting that the agglomeration of PML-NBs was not an irreversible process. Neither As$^{3+}$-induced solubility shift nor SUMOylation of PML was irreversible phenomena, because removal of As$^{3+}$ from the culture medium reverted those As$^{3+}$-induced biochemical changes in CHOGFPPML cells (Fig 5B) and in HEKGFPPML cells (S4 Fig).

## SUMOylation and ubiquitination of PML

PML proteins are known to be modified by SUMOylation and subsequent ubiquitination by RNF4 E3 ubiquitin ligase in As$^{3+}$-exposed cells [21, 34]. We investigated whether post-translational modification of GFPPML with SUMO and ubiquitin molecules occurs while GFPPMLs are still soluble in cold RIPA. Fig 6 shows that exposure to As$^{3+}$ converted GFPPML from the RIPA-soluble to the RIPA-insoluble form, and SUMOylated GFPPML remained in the RIPA-

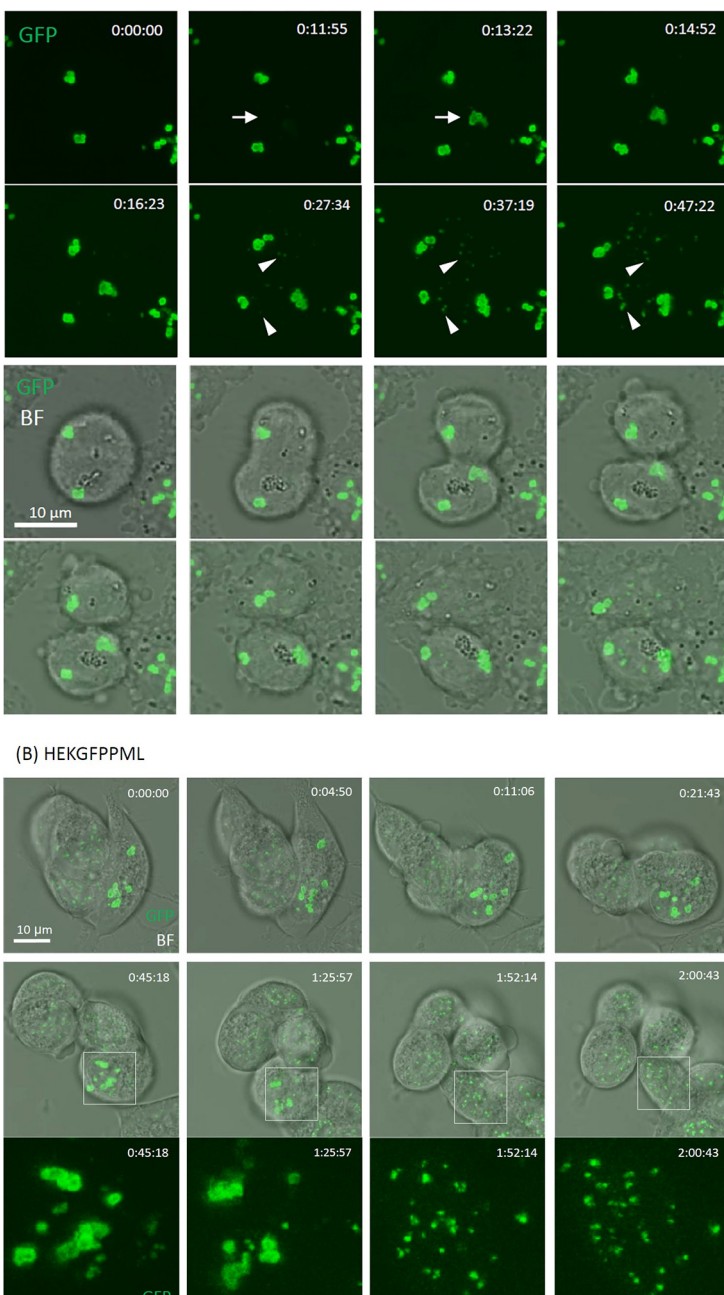

**Fig 2.** Topological changes of PML–NBs and peri–nuclear PML aggregates in CHOGFPPML (A) and HEKGFPPML cells (B). GFP live images were obtained by confocal laser scanning microscopy in z–stacking mode. (A) Time–lapsed images of CHOGFPPML cells were captured during cell division and spreading. The upper 8 panels show GFP images, and the lower 8 panels show overlaid images with the corresponding bright field (BF). The time counter is shown in the right upper corner of each GFP panel. Arrows show *de novo* appearance of a large peri–nuclear PML aggregate. The accretion of small GFP speckles began at 0:11:55 and the large toroidal PML aggregate was formed clearly after 1 min and 27 sec (at 0:13:22). Arrowheads show small nascent PML–NBs that appeared in daughter cells after cell division. (B) Time–lapsed images of HEKGFPPML cells were captured. The time counter is shown in the right upper corner of each panel. GFP fluorescence images in the boxed areas were enlarged and are shown in the bottom 4 panels. The toroidal PML–NBs were dissipated or dissolved (1:25:57), then were transformed into small granular dot–like PML–NBs (1:52:14) in the non–dividing cell.

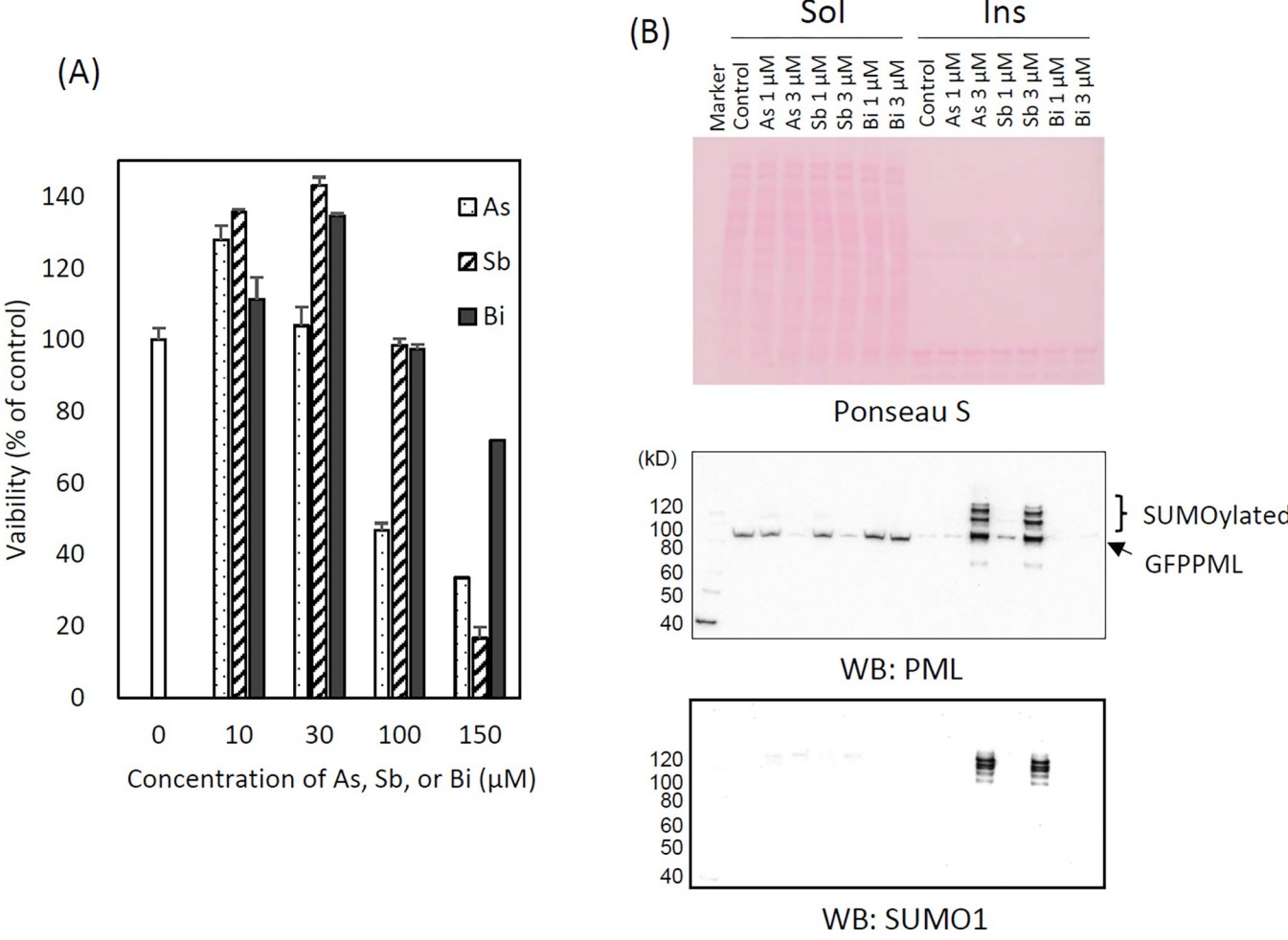

**Fig 3.** Changes in cell viability (A), and solubility shift and SUMOylation of PML (B) in response to trivalent arsenic (As), antimony (Sb), or bismuth (Bi) ions in CHOGFPPML cells. (A) The cells were exposed to various concentrations of $As^{3+}$, $Sb^{3+}$, or $Bi^{3+}$ for 24 h. $As^{3+}$ was significantly more cytotoxic than $Sb^{3+}$ and $Bi^{3+}$ in CHOGFPPML cells as analyzed by two–way ANOVA followed by Tukey's multiple comparison. Data are presented as means ± SEM of quadruplicated wells. (B) The solubility of PML in cold RIPA buffer decreased in response to 2 h–exposure to 1–3 μM $As^{3+}$ or $Sb^{3+}$. GFPPML was SUMOylated (SUMO1) following exposure to either 3 μM $As^{3+}$ or $Sb^{3+}$. These biochemical changes were not observed in $Bi^{3+}$–exposed cells. The major cold RIPA–insoluble proteins observed by Ponseau S–staining are histones. 'Sol' and 'Ins' denote the RIPA–soluble and–insoluble fractions, respectively.

soluble fraction at 1 h post-exposure in both CHOGFPPML (Fig 6A) and HEKGFPPML cells (Fig 6B). Accordingly, immunoprecipitation was performed using the RIPA-soluble fraction with magnetic bead-tagged anti-GFP antibody using the clear supernatants of RIPA lysates obtained from untreated and $As^{3+}$-exposed (1 h) CHOGFPPML cells (Fig 7A) and HEKGFPPML cells (Fig 7B). Modification on GFPPML with SUMO2/3 and SUMO1, and ubiquitination, were clearly observed in these cells after 1 h-exposure to $As^{3+}$. It should be noted that the amount of SUMOylated GFPPML immunoprecipitated in the untreated cells was less than that of $As^{3+}$-exposed cells.

## Effects of SUMOylation and ubiquitination inhibitors on PML-NBs

Recently, TAK243 (an inhibitor of UBA1-Ub thioester bond formation) and ML792 (an inhibitor of UBA2-SUMO thioester bond formation) were shown to effectively inhibit the ubiquitination and SUMOylation of PML, respectively [35, 36]. Exposure to 20 μM ML792 for up to 8

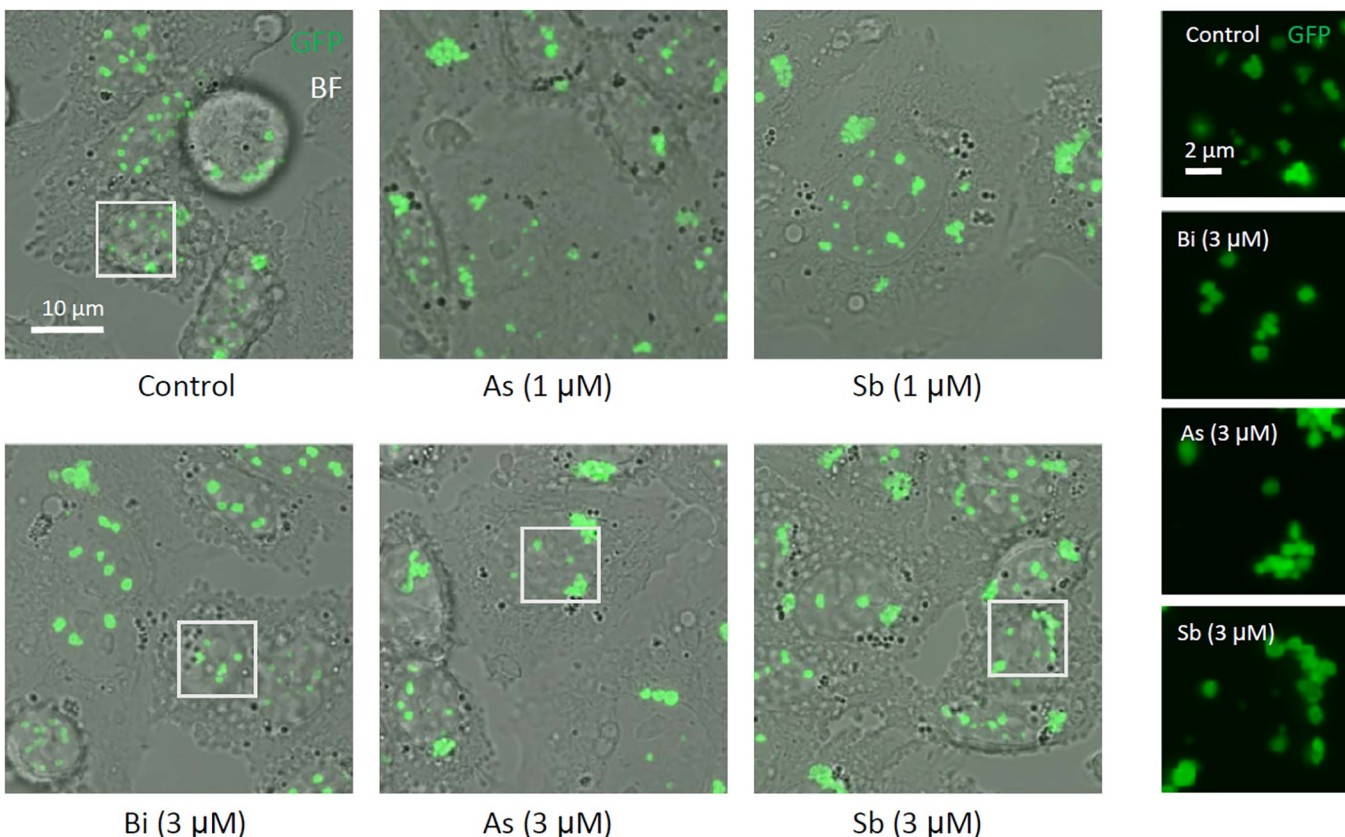

**Fig 4. Agglomeration of PML–NBs in response to As$^{3+}$ or Sb$^{3+}$ in CHOGFPPML cells.** The cells were exposed to 1 or 3 μM As$^{3+}$, Sb$^{3+}$, or Bi$^{3+}$ for up to 24 h. Contrary to the dramatic solubility change and SUMOylation of PML in response to As$^{3+}$ and Sb$^{3+}$, the intranuclear distribution of PML–NBs was not explicitly changed by 2 h–exposure to As$^{3+}$ or Sb$^{3+}$. After exposure to As$^{3+}$ or Sb$^{3+}$ (1 and 3 μM) for 24 h, however, agglomeration of PML–NBs was observed, while most PML–NBs were evenly distributed in Bi$^{3+}$–exposed and untreated cells. GFP images within the white boxes were enlarged and are shown in the right panels.

h did not significantly change the viability of HEKGFPPML cells, although longer exposure slightly reduced the number of viable cells compared to untreated cells (S5A Fig). TAK243 was cytotoxic at a concentration of 10 μM and cells started to shrink after 5 h of culture with TAK243. It is noteworthy that ML792 reduced the cytotoxicity of TAK243 (S5B Fig), suggesting that SUMOylation may enhance the adverse effects caused by the accumulation of ubiquitin-free and improperly folded proteins.

Inhibiting SUMOylation by ML792 reduced the number of PML-NBs and reciprocally increased their sizes in both CHOGFPPML (Fig 8A) and HEKGFPPML cells (Fig 8B), although the size difference was not statistically significant in CHOGFPPML cells. The number of PML-NBs was slightly reduced by TAK243 in CHOGFPPML cells but not in HEKGFPPML cells. These results indicated that the SUMOylation of PML is important to maintain the number and size of PML-NBs.

Fig 9 shows that TAK243 completely inhibited the ubiquitination of RIPA-soluble proteins in both CHOGFPPML (Fig 9A) and HEKGFPPML cells (Fig 9B). In the RIPA-insoluble fraction, the major anti-multi-ubiquitin reactive protein (24 kD) that disappeared upon TAK243 treatment is probably mono-ubiquitinated H2A and/or H2B [37]. SUMOylation of GFPPML with either SUMO1 or SUMO2/3 was inhibited by ML792. SUMO2/3 monomers, present in the RIPA-soluble fraction, essentially disappeared in CHOGFPPML (Fig 9A) and were

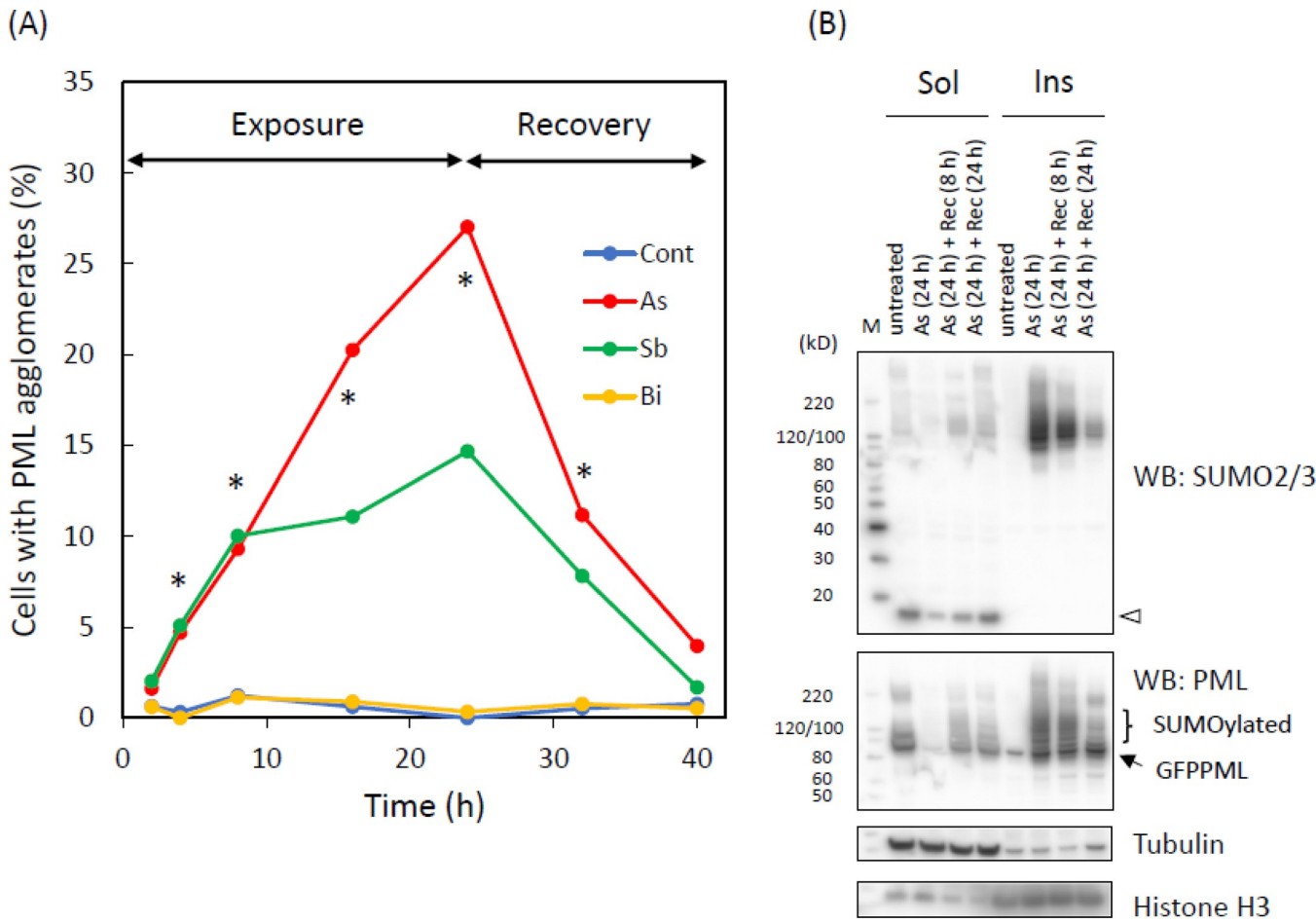

**Fig 5.** Changes in the percentage of cells with agglomerated PML–NBs (A), and recovery of PML in the RIPA–soluble fraction (Sol) from a 24 h–exposure to As³⁺ in CHOGFPPML cells (B). (A) The cells were exposed to 3 μM As³⁺, Sb³⁺, or Bi³⁺ for 24 h, and further cultured in fresh medium for 16 h. The numbers of cells with the agglomerated PML–NBs were counted at 2, 4, 8, 16, 24, 32 (8 h in recovery), and 40 h (16 h in recovery). At least 300 cells of each group were examined for the presence of agglomerated PML–NBs. The agglomeration was defined as five or more than five PML–NBs in an aggregated form. *, Significantly different among the groups as examined by $\chi^2$ test. (B) The cells were first exposed to 3 μM As³⁺ for 24 h. The cells were lysed with the RIPA buffer immediately or washed and further cultured in As³⁺–free culture medium for 8 or 24 h before lysis. Lane 1 and 5, untreated; Lane 2 and 6, 24–h exposure to As³⁺; Lane 3 and 7, 24–h exposure to As³⁺and 8–h recovery in As³⁺–free culture medium; Lane 4 and 8, 24–h exposure to As³⁺ and 24–h recovery in As³⁺–free culture medium. The unconjugated GFPPML and GFPPML conjugated with SUMO2/3 in the RIPA–insoluble fraction (Ins) decreased during the culture in As³⁺–free culture medium. An open arrowhead indicates SUMO2/3 monomers.

reduced in HEKGFPPML cells (Fig 9B) after 2 h-exposure to As³⁺. However, the amount of SUMO2/3 monomers was not changed by As³⁺ in ML792-pre-treated cells. Exposure to As³⁺ caused the solubility shift and SUMOylation of GFPPML. Although SUMOylation was inhibited by ML792, a substantial amount of non-SUMOylated GFPPML was recovered in the RIPA-insoluble fraction after exposure to As³⁺, suggesting that SUMOylation is not the cause but rather the consequence of the solubility change of GFPPML. Alternatively, the solubility shift and SUMOylation of GFPPML may occur independently in response to As³⁺. UBA2, a SUMO-activating enzyme, was not detected in the RIPA-insoluble fraction, regardless of exposure to As³⁺.

Finally, we investigated the intracellular distribution of SUMO2/3 when SUMOylation was inhibited. CHOGFPPML cells were pre-cultured in the presence or absence of ML792 and exposed to As³⁺ or left untreated for 2 h before immunostaining with anti-SUMO2/3. It is of

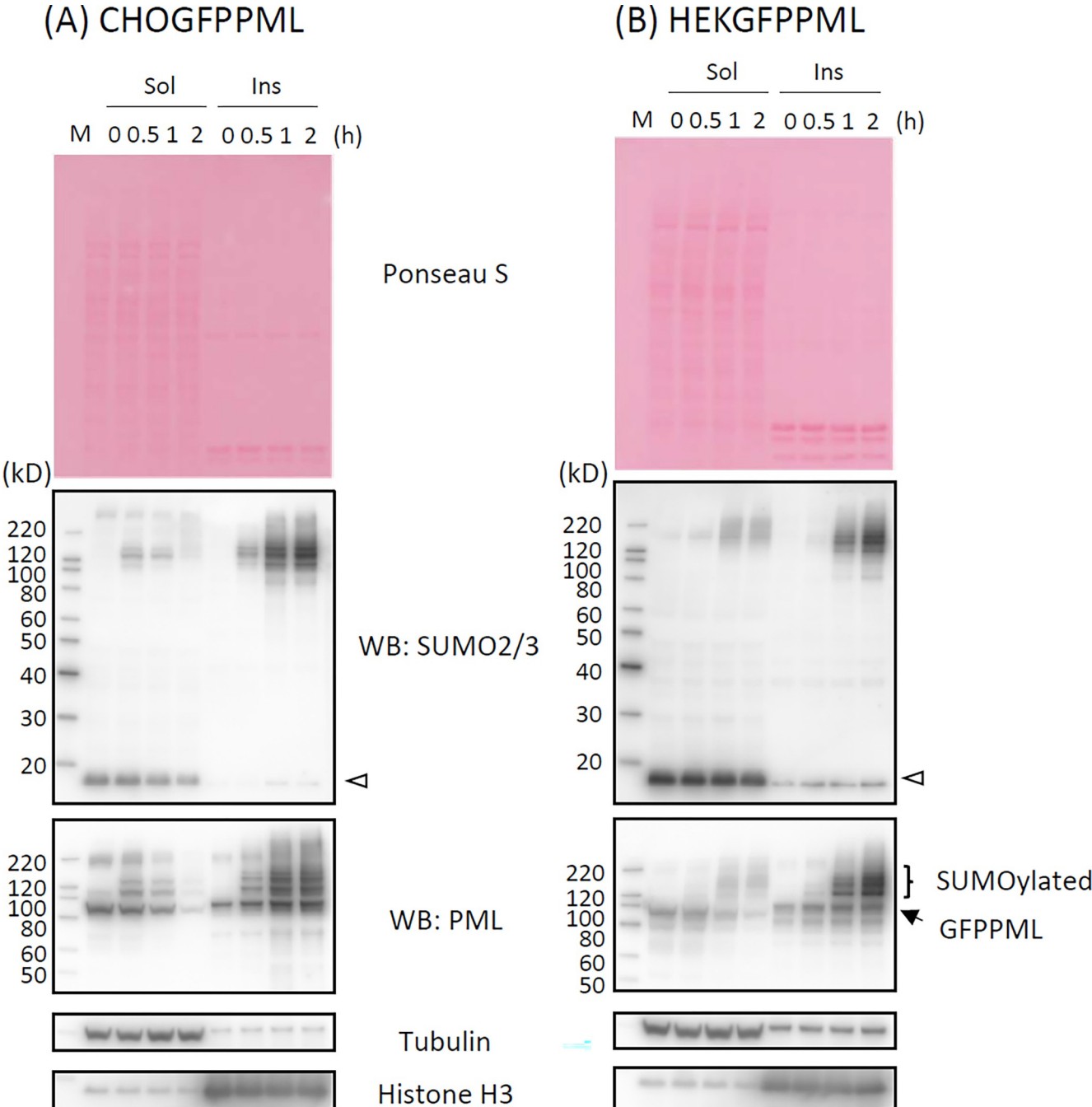

**Fig 6.** Time–course changes in the solubility and SUMOylation of PML following exposure to 3 μM As$^{3+}$ in CHOGFPPML (A) and HEKGFPPML cells (B). The cells were exposed to 3 μM As$^{3+}$ for 0 (untreated), 0.5, 1, and 2 h. SUMOylation of GFPPML was observed in the RIPA–soluble fraction (Sol) after 0.5 h–exposure. The solubility of unmodified GFPPML decreased after 1 h, and most of the GFPPML and SUMOylated GFPPML was recovered in the RIPA–insoluble fraction (Ins) after 2 h–exposure to As$^{3+}$. An open arrowhead indicates SUMO2/3 monomers.

interest to note that the ratio of SUMO2/3 to PML of the peri-nuclear PML aggregates was lower than that of PML-NBs regardless of As$^{3+}$-exposure in the absence of ML792 as judged by the relative fluorescence intensities (arrows, Fig 10). In contrast, the SUMO/PML ratio was almost the same between peri-nuclear PML aggregates and PML-NBs when the cells were pre-

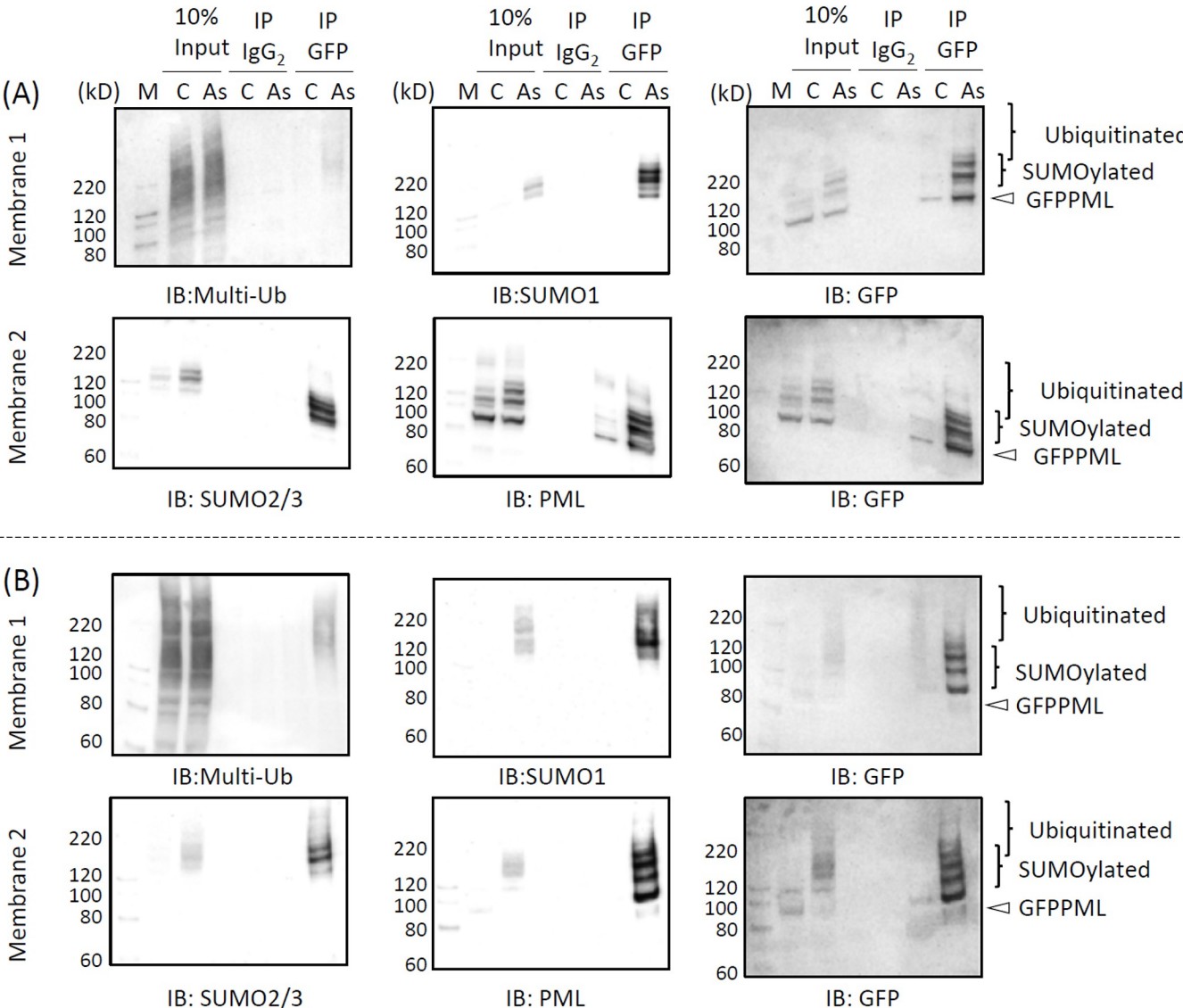

**Fig 7.** Immunoprecipitation analyses to detect the SUMOylation and ubiquitination of PML in CHOGFPPML (A) and HEKGFPPML cells (B). The cells were exposed to 3 µM As$^{3+}$ (As) or left untreated (C) for 1 h and the RIPA–soluble fractions were collected. The samples were centrifuged again to obtained clear supernatants and subjected to immunoprecipitation with anti–GFP antibody–magnetic beads. The immunoprecipitates were resolved by SDS–PAGE and subsequent western blotting. The membranes were sequentially probed with anti–multi-Ub, anti–SUMO1, and anti–GFP antibodies (membrane 1) or with anti–SUMO2/3, anti–PML, and anti–GFP antibodies (membrane 2).

treated with ML792, suggesting that ML792 disturbs the intranuclear restoration of SUMO molecules.

## Discussion

### PML-NBs and peri-nuclear PML aggregates

Most PML-NBs occurred in the toroidal shape in CHOGFPPML cells and in the granular dot-like shape in HEKGFPPML cells (Figs 1 and 2). In addition to PML-NBs, peri-nuclear PML aggregates appeared to be accretions of several toroidal PML bodies in CHOGFPPML cells. These observations suggest that the CHO cells have a propensity for toroidal PML-NBs,

(A) CHOGFPPML

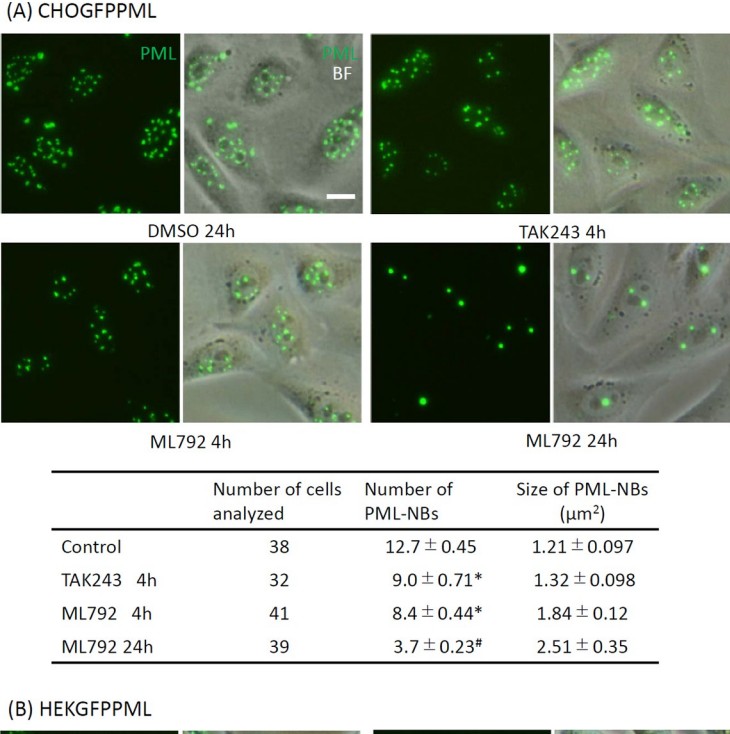

| | Number of cells analyzed | Number of PML-NBs | Size of PML-NBs ($\mu m^2$) |
|---|---|---|---|
| Control | 38 | $12.7 \pm 0.45$ | $1.21 \pm 0.097$ |
| TAK243 4h | 32 | $9.0 \pm 0.71$* | $1.32 \pm 0.098$ |
| ML792 4h | 41 | $8.4 \pm 0.44$* | $1.84 \pm 0.12$ |
| ML792 24h | 39 | $3.7 \pm 0.23$# | $2.51 \pm 0.35$ |

(B) HEKGFPPML

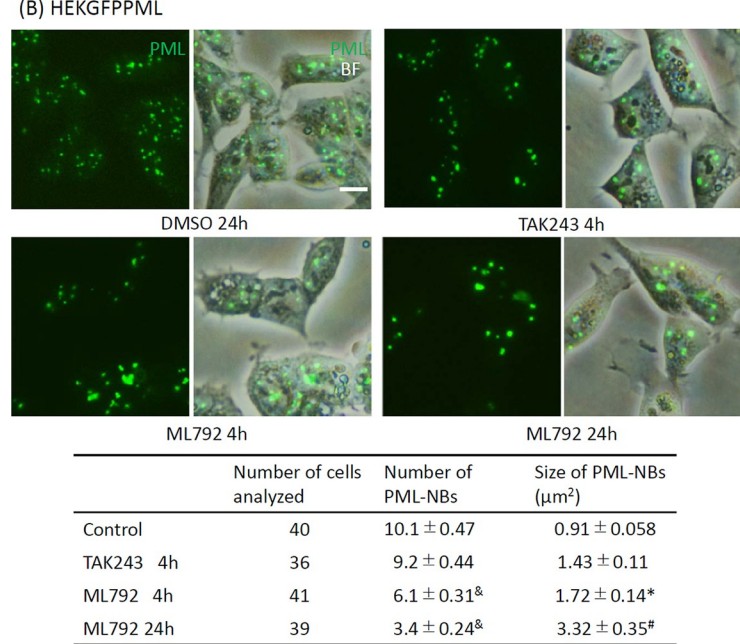

| | Number of cells analyzed | Number of PML-NBs | Size of PML-NBs ($\mu m^2$) |
|---|---|---|---|
| Control | 40 | $10.1 \pm 0.47$ | $0.91 \pm 0.058$ |
| TAK243 4h | 36 | $9.2 \pm 0.44$ | $1.43 \pm 0.11$ |
| ML792 4h | 41 | $6.1 \pm 0.31$& | $1.72 \pm 0.14$* |
| ML792 24h | 39 | $3.4 \pm 0.24$& | $3.32 \pm 0.35$# |

**Fig 8.** Effects of TAK243 (ubiquitination inhibitor) and ML792 (SUMOylation inhibitor) on the number and size of PML–NBs in CHOGFPPML (A) and HEKGFPPML cells (B). The cells were cultured in the presence of 0.1% DMSO (control), 10 μM TAK243 for 4 h, or 20 μM ML792 for 4 and 24 h. The number of PML–NBs in both cell types decreased and their sizes increased significantly by treatment with ML792. TAK243 was less effective than ML792 in decreasing the number of PML–NBs, and the cells looked shrunken and slightly damaged after 6 h of culture with 10 μM TAK243. Data are presented as the mean ± SEM. *, Significantly different from control group. #, Significantly different from all other groups. &, Significantly different from control and TAK243 groups. Scale bar = 10 μm.

although it is not clear why the annular condensation of GFPPML occurs in CHO cells. Given that SUMOylation of GFPPML proceeded normally in both cell types, PML-NBs may need some other scaffold or client molecules to form stable toroidal shapes. The disassembly of large

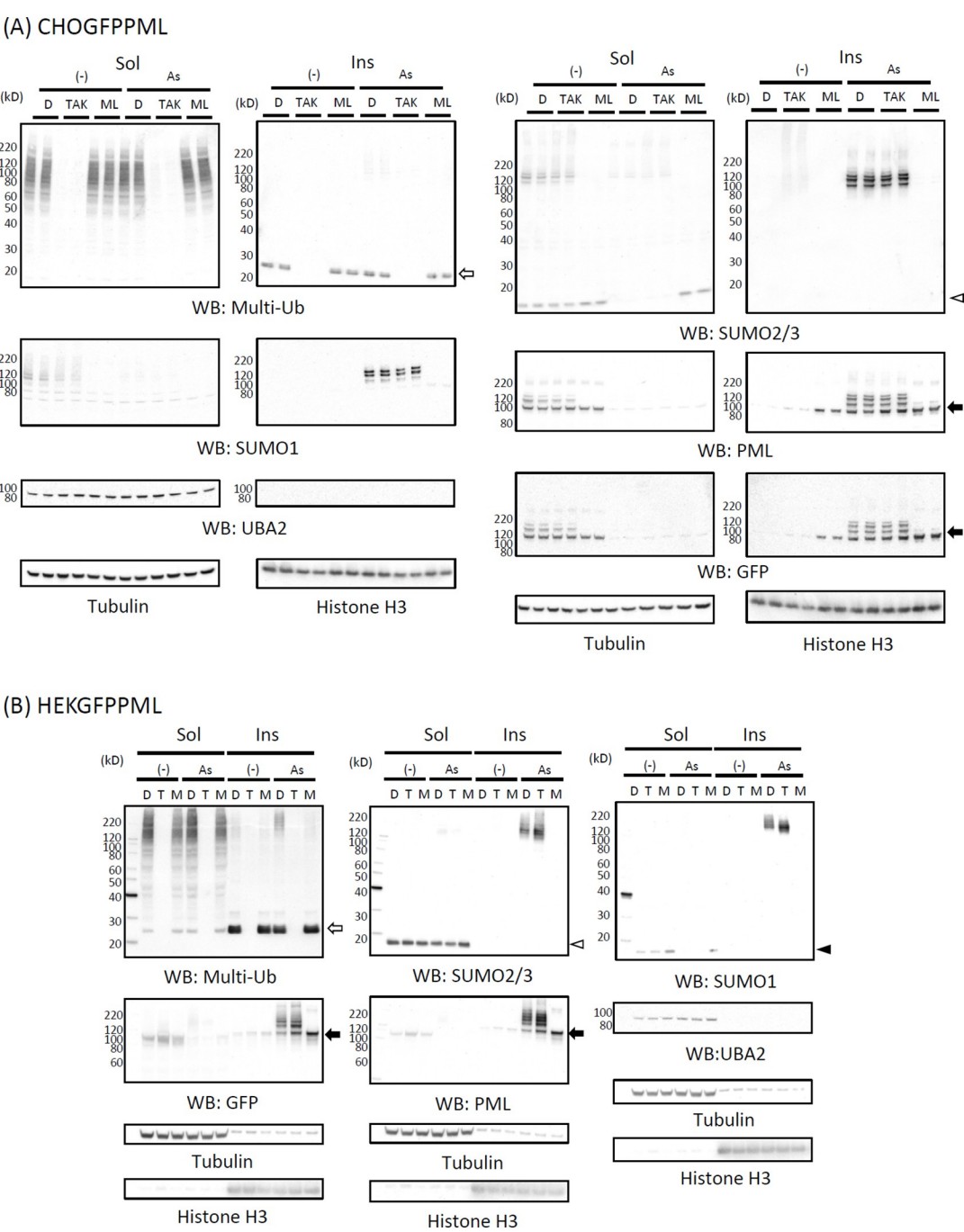

**Fig 9.** Effects of TAK243 and ML792 on As[3+]–induced biochemical changes of PML in CHOGFPPML (A) and HEKGFPPML cells (B). The cells were pre–treated with 0.1% DMSO, 10 μM TAK243, or 20 μM ML792 for 3 h, then further treated with 3 μM As[3+] or left untreated for 2 h. The RIPA–soluble and–insoluble fractions were prepared as described previously, and the proteins were resolved by SDS–PAGE followed by western blotting. For CHOGFPPML cells, samples following each treatment were obtained from 2 separate wells. The membranes were probed sequentially with the indicated antibodies, then reprobed with a mixture of HRP–tagged anti–tubulin and HRP–tagged anti–histone H3 antibodies. Open arrow, mono–ubiquitinated histone; Closed arrow, GFPPML; Open arrowhead, SUMO2/3 monomers; Closed arrowhead, SUMO1 monomers.

toroidal PML-NBs into small granular dot-like PML-NBs in the HEKGFPPML cell nucleus (Fig 2B) indicates that PML-NBs are not rigid aggregates, and that toroidal PML-NBs in HEK cells are more labile than those in CHO cells. Non-spherical PML-NBs are prone to forming

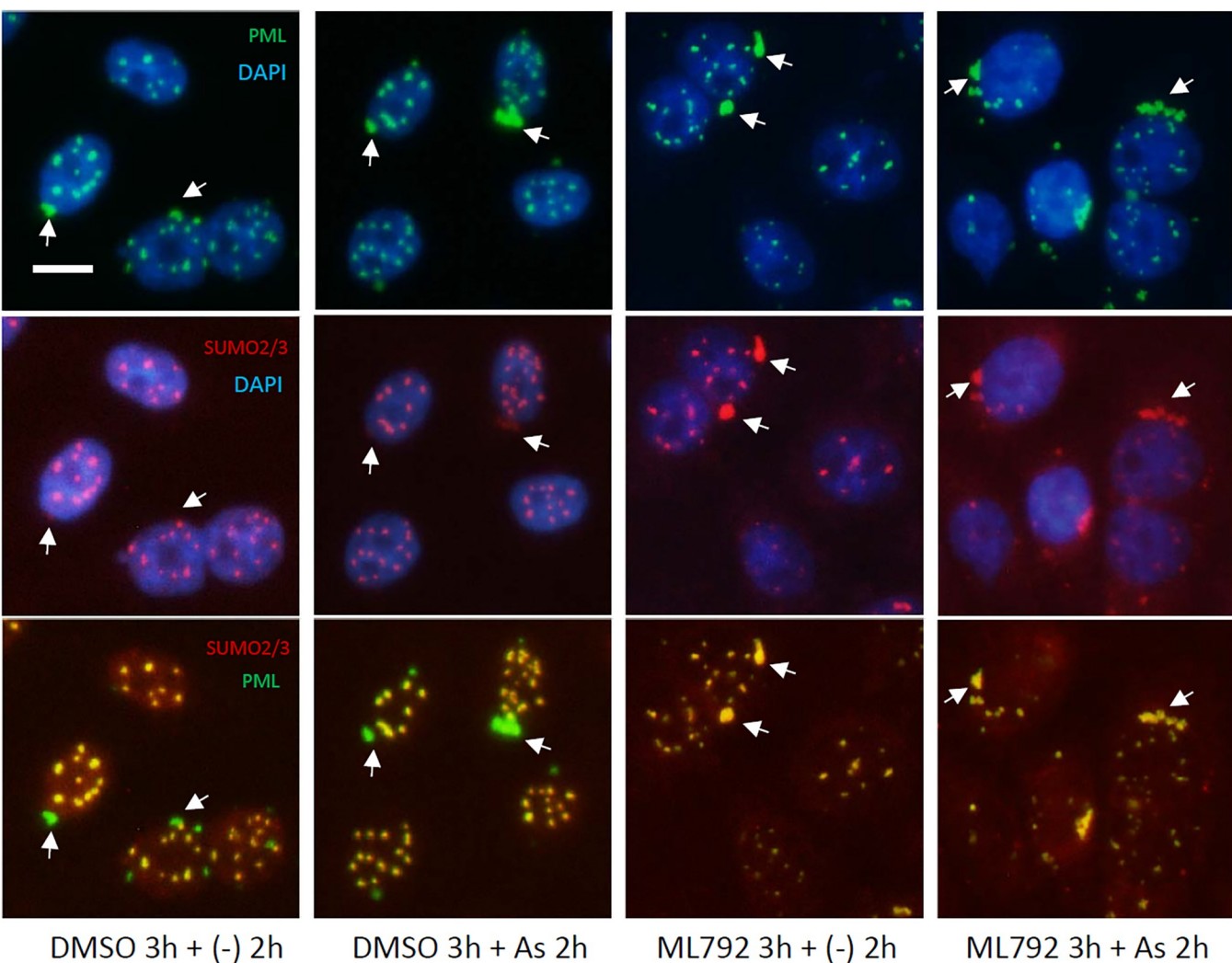

**Fig 10. Fluorescent immunostaining of ML792–pretrreated CHOGFPPML cells to detect PML and SUMO2/3.** The cells were treated with 0.1% DMSO or 20 µM ML792 for 3 h, then further treated with 3 µM As$^{3+}$ or left untreated for 2 h. Note that SUMO2/3 in peri–nuclear PML aggregates (arrow) was decreased in the DMSO–treated cells, whereas a substantial amount of SUMO2/3 retained with GFPPML in the ML792–treated cells. Scale bar = 10 µm.

hydrogels with less fluidity in the sol-gel balance of non-membrane bodies because a liquid-like nature would make these bodies spherical due to viscous relaxation [38]. Our current finding that toroidal PML-NBs were converted into small granular dot-like PML-NBs in the HEKGFPPML cell nucleus may shed light on the rheological nature of PML-NBs.

A systematic regulatory network deciphering study indicated that most cells with increased PML-NBs contain S phase DNA and low 5-ethynyl-2'-deoxyuridine (EdU) incorporation, suggesting that increased PML-NBs correlate with decreased DNA replication rates [39]. It has been reported that the number of PML-NBs increases in early S phase in SK-N-SH and *GFP-PML-IV* transduced U-2OS cells due to fission of the parent NBs and not to increased PML protein levels [10]. Although the transformation of large toroidal PML-NBs into small granular dot-like ones in HEKGFPPML cells (Fig 2B) appeared to begin with dissolution or dissipation rather than fission in the present study, the intranuclear dynamics of PML-NBs may be fine-tuned with DNA synthesis.

## Effects of As3+, Sb3+, and Bi3+ on PML

One of the most notable biochemical changes in $As^{3+}$-exposed cells was the solubility shift from the cold RIPA-soluble to the RIPA-insoluble form, and SUMOylation of PML [40, 41]. These changes were also observed in $Sb^{3+}$-exposed but not in $Bi^{3+}$-exposed GFPPML-expressing cells (Fig 3), which is consistent with findings using other cell types [13, 42]. $Bi^{3+}$ forms a stable complex with metallothionein in the cytosol and might not efficiently reach the nucleus [43]. The solubility shift and SUMOylation of PML appear to be induced by TGF-β, as well as by $As^{3+}$ or $Sb^{3+}$ [44]. The PML protein level, and the number and size of PML-NBs, are increased by IFNα in BL41 cells; however, the solubility change of PML does not occur in IFNα-treated cells. In contrast, TGF-β disrupts PML-NBs and changes PML from the RIPA-soluble (nucleoplasm) to the RIPA-insoluble form (nuclear matrix), and further enhances the SUMOylation of PML. Accordingly, a large amount of SUMOylated PML was found in the RIPA-insoluble fraction in TGF-β-treated and IFNα-pre-treated cells [44]. However, the involvement of cytokines in $As^{3+}$-exposed cells remains to be elucidated. Intriguingly, prolonged exposure to $As^{3+}$ caused agglomeration of PML-NBs which was reversible after removal of $As^{3+}$ from the culture medium (Figs 4 and 5). To our best knowledge, this is the first quantitative study to report the agglomeration of PML-NBs. Since the agglomeration of PML-NBs was observed 4 h after exposure to $As^{3+}$, either solubility change or SUMOylation of PML or both may lead to the agglomeration.

## A role of SUMOylation in PML-NBs

MAPPs are aggregated forms of PML that appear after nuclear membrane breakdown, and CyPNs are formed in the peri-nuclear regions of daughter cells after mitosis [45]. MAPPs and CyPNs appear to be devoid of SUMO, Sp100, and Daxx, which are well-known PML-NB client proteins [46, 47]. We did not differentiate MAPPs and CyPNs, referring to these PML extranuclear PML bodies as peri-nuclear PML aggregates in the present study. Although the biological functions of these peri-nuclear PML aggregates have not been elucidated [48], the aggregates can be used as markers of laminopathies [49]. Our findings that PML colocalized with SUMO2/3 in PML-NBs and peri-nuclear PML aggregates contained less SUMO2/3 are consistent with these previous studies. SUMO2/3 distributed almost evenly between PML-NBs and peri-nuclear PML aggregates in the presence of ML792, an inhibitor of SUMO-activating enzymes, irrespective of $As^{3+}$ exposure (Fig 10). Since SUMOylation of GFPPML was almost completely inhibited by ML792 (Fig 9), it is plausible that SUMO monomers non-covalently associate with peri-nuclear PML aggregates as well as with PML-NBs in the presence of ML792.

The 69-kDa isoform of PML mutated at the B-box (B1C17C20AΔAla, B1C25C28AΔAla, and B2C21C24ΔAla) does not form PML-NBs and is instead present diffusely over the nucleus [11]. The three main SUMOylation sites (K65, 160, and 490) and the SIM are not essential for formation of the PML multimer lattice or the outer shell of PML-NBs. SUMO2/3 appears to be more closely associated with PML compared to other partner proteins such as Sp100, Daxx, SUMO1, and RNF4 [50]. In addition, depletion of SUMO3 reduces the number of PML-NBs [51]. Recently, PML-specific sequences (F52QF54 and L73) were shown to play essential roles in PML-NB assembly, recruitment of Ubc9, and SUMOylation [16]. The depletion of SUMO-specific protease 1 (SUSP1/SENP6), which dismantles highly conjugated SUMO2/3 proteins, caused marked accumulation of SUMO2/3 in PML-NBs [52] and increased the average number of PML-NBs per nucleus from 3.7 to 5.9 in HeLa cells [53]. Thus, the balance of SUMOylation and deSUMOylation of PML is critical for the appropriate maintenance of PML-NBs. In addition to SUMOylation and deSUMOylation of PML, PML-NBs might be regulated by non-

PML-NB-associated proteins. The expression of either Ki-1/57 or CGI-55 proteins, which have three SUMOylation sites, reduces the number of PML-NBs in both untreated and As$^{3+}$-treated HeLa cells, whereas the expression of Ki-1/57 mutant (K/3R) reduces the number of PML-NBs in untreated cells but not in As$^{3+}$-treated cells [54]. The inhibition of SUMOylation by ML792 increased the size of PML-NBs in the present study (Fig 8), suggesting that the SUMOylation indeed regulates the scaffold structure of PML-NBs.

Blocking ubiquitination using TAK243 resulted in the build-up of SUMOylated proteins in HeLa and U-2OS cells, with PML-NBs apparently functioning as primary accumulation sites for newly synthesized and SUMOylated proteins [36]. Ubiquitination by RNF4 E3 ligase via the reciprocal SUMO-SIM interaction is proposed to be the main cascade for the proteasomal degradation of SUMOylated PML-RARα [8, 19, 55]. PML-NBs may be transiently employed to store harmful SUMOylated proteins by phase separation before proteolysis by the ubiquitin-proteasome system [56]. However, functional roles of SUMOylated PML remain elusive, because modification with SUMO protects IκBα [57] and Glis2/NPHP7, a Cys2/His2 zinc finger transcription factor [58], from ubiquitin-proteasome degradation by competing the same lysine [3]. An ancillary but interesting finding in the present study is that the cytotoxic effect of TAK243 was antagonized by ML792 (S5 Fig), suggesting that SUMOylation may be harmful to the cells when ubiquitination is inhibited.

## Conclusions

In summary, topological features of PML-NBs were different between CHO and HEK cells, although biochemical responses of PML to As$^{3+}$ such as solubility shift and SUMOylation were almost the same between these two different cells. The solubility shift of PML occurred prior to SUMOylation. The number, size, and shape of PML-NBs were not changed by 2-h exposure to As$^{3+}$. However, the prolonged exposure to As$^{3+}$ or Sb$^{3+}$ caused agglomeration of PML-NBs which is reversed after removal of these metalloids from the culture medium. The present results warrant the importance of SUMO-dependent and -independent dynamic nature of PML-NBs which are implicated in phase separation cell biology and human diseases.

## Supporting information

**S1 Fig. Emergence and uneven partitioning of peri-nuclear PML aggregates in dividing untreated CHOGFPPML cells.** Live GFP images were captured during cell division by confocal laser scanning microscopy in z-stacking mode. The upper and middle 8 panels show GFP alone and the corresponding GFP-bright field overlaid images, respectively. The lower 8 panels show enlarged and intensified GFP images of a peri-nuclear PML aggregate indicated with arrows on the top four panels with four additional time-lapsed GFP images. The time counter is shown in the right upper corner of each panel. The peri-nuclear PML aggregates appear to be comprised of 2–4 toroids. The nascent small PML-NBs (arrowheads) appeared as the daughter cells spread.
(PDF)

**S2 Fig. Uneven partitioning of peri-nuclear PML aggregates in As$^{3+}$-exposed CHOGFPPML cells.** Image capture started 30 min after the addition of As$^{3+}$. The upper and lower 10 panels show GFP images alone and the corresponding GFP-bright field overlaid images, respectively. The nascent small PML-NBs (arrowheads) appeared as the daughter cells spread. See also the legend to **S1 Fig.**
(PDF)

**S3 Fig. Inhibition of proteasomal activity by As$^{3+}$, Sb$^{3+}$, Bi$^{3+}$, and Cd$^{2+}$.** The proteasomal inhibitory effect was assayed using a 20S proteasome assay kit. Briefly, the fluorescence (Ex/Em, 360/460 nm) of the reaction mixture (proteasome, Suc-LLVY-AMC, and SDS) was monitored every 1 min in the presence or absence of As$^{3+}$, Sb$^{3+}$, Bi$^{3+}$, Cd$^{2+}$, or epoxomicin (a positive control) at 30°C for 20 min using a microplate reader (Infinite M Plex, TECAN, Männedorf, Switzerland). As$^{3+}$, Sb$^{3+}$, Bi$^{3+}$, and Cd$^{2+}$ were added to the reaction mixture at a final concentration of 30 μM. The fluorescence was increased linearly and the increment of fluorescence in 20 min was used for evaluation of relative inhibitory activity of proteasome. Data are presented as means ± SEM of 4 replicate measurements.
(PDF)

**S4 Fig. Recovery of SUMO2/3 and GFPPML proteins in the RIPA-soluble fraction (Sol) from 24 h-exposure to As$^{3+}$ in HEKGFPPML cells.** The cells were exposed to 3 μM As$^{3+}$ for 24 h or left untreated. The As$^{3+}$-exposed cells were lysed with the RIPA buffer immediately or washed and further cultured in As$^{3+}$-free culture medium further for 8 or 24 h. 1, untreated; 2, 24 h exposure to As$^{3+}$; 3, 24 h exposure to As$^{3+}$ and 8 h recovery in As$^{3+}$-free culture medium; 4, 24 h exposure to As$^{3+}$ and 24 h recovery in As$^{3+}$-free culture medium. An open arrowhead indicates SUMO2/3 monomers. The unconjugated GFPPML and GFPPML conjugated with SUMO2/3 in the RIPA-insoluble fraction (Ins) decreased during the culture in As$^{3+}$-free culture medium.
(PDF)

**S5 Fig.** Time-course changes in the number of viable ML792-exposed cells (A) and the inhibitory effects of ML792 on the cytotoxicity of TAK243 (B) in HEKGFPPML cells. (A) The cells were exposed to 0.1% DMSO (open columns) or 20 μM ML792 (hatched columns) for 2, 4, 8, 15 h. The number of viable cells was assayed using WST-8 agent. Data are presented as means ± SEM of five wells. (B) The cells were exposed to 0, 2.5, 5, and 10 μM TAK243 for 24 h in the presence or absence of 20 μM ML792. Data were analyzed by two-way ANOVA followed by Tukey's multiple comparison. The viability of ML792-treated cells was significantly elevated compared to 0.1% DMSO-treated cells in the presence of 10 μM TAK243.
(PDF)

**S1 Raw images.**
(PDF)

**S1 Data.**
(XLSX)

## Acknowledgments

The authors would like to thank Dr. Ayaka Kato-Udagawa and Ms. Mihoko Tadano for their technical assistance.

## Author Contributions

**Conceptualization:** Seishiro Hirano.

**Data curation:** Seishiro Hirano.

**Formal analysis:** Seishiro Hirano.

**Funding acquisition:** Seishiro Hirano.

**Investigation:** Seishiro Hirano, Osamu Udagawa.

**Methodology:** Seishiro Hirano, Osamu Udagawa.

**Project administration:** Seishiro Hirano.

**Writing – original draft:** Seishiro Hirano.

**Writing – review & editing:** Seishiro Hirano, Osamu Udagawa.

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
