## [Decision Letter · Decision Letter 0]

30 Dec 2021

PONE-D-21-36000Effects of arsenic on the topology and solubility of promyelocytic leukemia (PML)-nuclear bodiesPLOS ONE

Dear Dr. Hirano,

Thank you for submitting your manuscript to PLOS ONE. After careful consideration, we feel that it has merit but does not fully meet PLOS ONE’s publication criteria as it currently stands. Therefore, we invite you to submit a revised version of the manuscript that addresses all the points raised during the review process.

The experiments are well conducted but the originality and significance of the data can be better presented and exploited.

I invite you to rewrite the manuscript to better highlight the biological questions addressed and the original results obtained. The manuscript also needs English proofreading to make the style clearer

Please improve the quality of microscopy images.

Please define more precisely the different types of bodies they mention (toroid bodies, annular bodies, amorphous bodies, small dots, …) and provide metrics to characterize these bodies and their dynamics.

Please extend the discussion to better explain the significance of your findings on the understanding of PML biology

We look forward to receiving your revised manuscript.

Kind regards,

Claude Prigent

Academic Editor

PLOS ONE

Journal Requirements:

The authors would like to thank Dr. Ayaka Kato-Udagawa and Ms. Mihoko Tadano for their technical assistance. This work was partially supported by a Grant-in-Aid from the Japan Society for the Promotion of Science (16K15386).

We note that you have provided funding information. However, funding information should not appear in the Acknowledgments section or other areas of your manuscript. We will only publish funding information present in the Funding Statement section of the online submission form. 

This work was partially supported by a Grant-in-Aid from the Japan Society for the Promotion of Science (16K15386 given to SH). The funders had no role in study design, data collection and analysis, decision to publish, or preparation of the manuscript.

Reviewers' comments:

Reviewer's Responses to Questions

**Comments to the Author**

1. Is the manuscript technically sound, and do the data support the conclusions?

Reviewer #1: Yes

Reviewer #2: Yes

2. Has the statistical analysis been performed appropriately and rigorously? 

Reviewer #1: Yes

Reviewer #2: N/A

3. Have the authors made all data underlying the findings in their manuscript fully available?

Reviewer #1: Yes

Reviewer #2: No

4. Is the manuscript presented in an intelligible fashion and written in standard English?

Reviewer #1: Yes

Reviewer #2: Yes

5. Review Comments to the Author

Reviewer #1: The authors analyze in the manuscript the effect of Arsenic on the solubility of the PML protein (isoform VI), using 2 different cell lines stably expressing exogenous, GFP-tagged PML-VI. They document that PML-nuclear bodies (PML-NBs) have different shapes and stability depending on the cells. They show that treatment of the cells for 2hrs with 3µM Arsenic (As3+) and also to as lesser extent antimony (Sb3+), but not bismuth (Bi3+), induces PML SUMOylation together with its relocalization from a soluble to an insoluble compartment, without affecting size, number and intranuclear distribution of PML NBs. However, upon prolonged (24 hrs) treatment, PML NBs agglomerated in the nucleus, in a reversible manner. The authors further show that upon As3+ treatment PML SUMOylation precedes its relocalization, but is not necessary for this relocalization. Rather, PML SUMOylation appears important to maintain the number and size of PML NBs.

General comments:

Overall, I find the manuscript technically sound, with data that support the conclusions presented by the authors. What remains unclear to me is the originality of the issues addressed in the manuscript, and consequently the novelty of the conclusions, as to my knowledge most of the information found in the manuscript has already been published previously. It seems to me that the interest of the manuscript could be greatly improved if the authors extensively rewrite it to better highlight the biological questions they want to address and the original results they obtained, while reducing the length given to what is already known.

Specific concerns:

- the authors analyze in the manuscript the isoform VI of PML, which is lacking a SUMO-interacting motif present in the other isoforms. Isn't this choice a concern when studying the solubility and the SUMOylation of the protein? I think this point should be discussed.

- the quality and resolution of the microscopy images is low (as judged from the pdf available for reviewing) and should be increased. Actually, it would be useful for many readers to display the exact contour of the nuclei, either by outlining them in the images of by adding photos of a DAPI (or equivalent) staining.

- I am not sure of the pertinence of testing the possible inhibition of the proteasome by As3+ and Sb3+ on 20S proteasome only. Indeed, one could imagine that these compounds inhibit more the 26S than the 20S proteasome, by interfering not directly with the proteolytic activities but with other activities or properties of the 26S proteasome. To strengthen this point, one possibility would be to better document that there is no accumulation of ubiquitylated proteins in cells upon these treatments, as indicated by the input fractions in Fig.7.

- if overall the quality of the English is adequate, the reading is sometimes difficult and it seems to me that the manuscript would benefit from editing by a native English speaker to make the style clearer. As it is, the meaning and therefore the conveyed message of some sentences is sometimes obscure. For example, may be I missed the point but I could not understand the sentence "suggesting that the topology was retained in the extranuclear region" in the first paragraph of the results.

- the use of the term 'life-time' in the introduction is not correct. The authors probably meant 'residence-time'?

- Fig 10 legend: SUMO2/3 is more (not less) abundant in peri-nuclear PML aggregates in ML792-treated cells

Reviewer #2: The promyelocytic leukemia (PML) protein is regulating multiple cellular activities (including DNA repair, apoptosis and senescence) and plays a critical role in oncogenesis and tumor progression. PML is sensitive to arsenic and regulated by SUMOylation and ubiquitylation. It is also well known to accumulate in nuclear bodies which are important for its activity. In this manuscript, the authors investigate the effect of arsenic and other metalloids on the accumulation of PML in nuclear bodies, its solubility and its SUMOylation and ubiquitylation. The results that are presented are solid are will certainly contribute to a better understanding of the interplay between arsenic treatment, PML posttranslational modifications, and PML bodies formation, although their novelty and significance is not obvious for non-specialists.

Major comments:

1) The authors have studied the topology of PML nuclear bodies and their dynamics in two cell lines. They have quantified the size and number of PML bodies, but most of their other observations on the shape of the bodies, remain qualitative, poorly documented and difficult to visualize in the figures that are provided. They should more precisely define the different types of bodies they mention (toroid bodies, annular bodies, amorphous bodies, small dots, …) and provide metrics to characterize these bodies and their dynamics.

2) The authors should complete their discussion and conclusion to better explain the significance of their findings on the understanding of PML biology.

Other comments:

3) Figure 1 and Figure S1: The quantification provided in Figure S1 is important and should be included in the main Figure rather than in the supplementary data.

4) The PLOS data policy requires authors to make all data underlying the findings described in their manuscript fully available. Thus, the individual data points used for the plots of Figures 3A, 4A, 8A, 8B, S1 S4 and S6 should be provided, either as scatter plots or in supplementary tables.

5) The legend of Figure 8 should state whether the indicated variations of PML nuclear bodies number and size are SD or SEM. Actually, I encourage the authors to use SD rather than SEM as summary statistics for this and all other figures as this enables the reader to evaluate data dispersion.

6) Figure annotations: The annotations of many figures could be improved to facilitate their understanding and interpretation. For instance, in Figure 1 the panel A and B represent similar experiments performed with two different cell lines (CHOGFPPML and EKGFPPML). The authors should include the names of these cell lines in the corresponding panels of this (and other) Figure(s). Similarly, in Figure 5B, the lanes 1,2,3 and 4 correspond to different time points of As3+ treatment. This could be explicitly notated in the Figure panel rather in the legend. These two examples are not exhaustive and I encourage the authors to think for each Figure panel whether more information could be provided to improve their legibility.

7) The authors should state the mechanism of action of TAK243 and ML792 in the main text.

6. PLOS authors have the option to publish the peer review history of their article (what does this mean?). If published, this will include your full peer review and any attached files.

Reviewer #1: **Yes: **Olivier Coux

Reviewer #2: No

---

## [Decision Letter · Decision Letter 1]

14 Mar 2022

PONE-D-21-36000R1Effects of arsenic on the topology and solubility of promyelocytic leukemia (PML)-nuclear bodiesPLOS ONE

Dear Dr. Hirano,

Thank you for submitting your manuscript to PLOS ONE. After careful consideration, we feel that it has merit but does not fully meet PLOS ONE’s publication criteria as it currently stands. Therefore, we invite you to submit a revised version of the manuscript that addresses the points raised during the review process.

As you will see from their comments, both reviewers consider the revisions to the manuscript, while they improve the manuscript, to be insufficient to allow publication. 

I am willing to propose a second round of minor revisions, but you will need to address all of their requests (see reviewer #1's list).

For reviewer #2, the definition and quantification of bodies still needs improvement. The text also needs to be improved as suggested.

Translated with www.DeepL.com/Translator (free version)

We look forward to receiving your revised manuscript.

Kind regards,

Claude Prigent

Academic Editor

PLOS ONE

Journal Requirements:

Reviewers' comments:

Reviewer's Responses to Questions

**Comments to the Author**

1. If the authors have adequately addressed your comments raised in a previous round of review and you feel that this manuscript is now acceptable for publication, you may indicate that here to bypass the “Comments to the Author” section, enter your conflict of interest statement in the “Confidential to Editor” section, and submit your "Accept" recommendation.

Reviewer #1: (No Response)

Reviewer #2: (No Response)

2. Is the manuscript technically sound, and do the data support the conclusions?

Reviewer #1: Yes

Reviewer #2: Yes

3. Has the statistical analysis been performed appropriately and rigorously? 

Reviewer #1: Yes

Reviewer #2: Yes

4. Have the authors made all data underlying the findings in their manuscript fully available?

Reviewer #1: Yes

Reviewer #2: Yes

5. Is the manuscript presented in an intelligible fashion and written in standard English?

Reviewer #1: Yes

Reviewer #2: Yes

6. Review Comments to the Author

Reviewer #1: The authors have addressed the concerns made by the 2 reviewers, but unfortunately in the minimal way: the changes mostly concern the text of the manuscript, and fail to really clarify the biological significance of the presented findings and to make the manuscript easier to read. Overall, I still find the manuscript weak, both in the solidity and in the originality of the conclusions that can be drawn from the experiments.

Having said that, the presented data globally support the conclusions proposed by the authors, and I think the manuscript could be published, depending of the opinion of the other reviewer and of the editor.

However, I thing that the following modifications, noted as I read the revised manuscript, should be made before acceptance for publication, as they will help to improve the readability:

> first paragraph: the residence-time of PML ranges  the residence-time of PML in PML bodies ranges...

> beginning of second paragraph: define B1 and B2 domains

> second paragraph: The unique subdomain appears to play a role in PML tetramer  tetramerization? Same sentence: the following subsequent PML-NB formation  remove either following or subsequent as both terms are redundant

-Material and methods: the real term is confoncal laser scanning microscopy and not confoncal laser microscopy

- Results: the definition of what the authors consider toroidal or irregular-shaped punctates is still unclear: this makes it difficult to visualize in the images the structures referred to in the text. Typically, I don't understand how the authors can conclude that in HEKGFPPML cells, the "occasionally found" toroidal PML-NB are those that dissipate in small PML-NBs.

> page 9, last sentence before Fig. 3 legend: ..: a possibility that that...

> Fig 5A: the m of agglomerates is missing in the vertical legend of the graph

>Fig 7: the authors must comment why there is less immunoprecipitated GFPPML from the soluble fraction of untreated cells than from that of As-treated cells. Shouldn't the opposite be expected according to the results shown in Fig.3?

> Fig 8: the meaning of the superscript symbols used in the tables should be explained

- Sup data:

> legend of S2 refers to the legend of S2

Reviewer #2: In their revised manuscript, the authors answered many questions of the reviewers. The implemented changes improved the overall quality of the manuscript. In particular, the authors now provide in a supplementary file the original data underlying most of the plots presented in the different figures of the manuscript. This is a significant improvement, not only because it is a requirement from the PLOS data policy, but also because it enables to assess data variability, which is not represented in the plots (the error bars represent SEM, not SD). The authors also improved the annotations of several figure so that they are now easier to understand. Other small qualitative improvements have also been properly implemented, as listed by the authors in their point-to-point answer.

Yet, in my opinion, two of the most critical concerns that I and the other reviewer had raised were not properly addressed:

1) I had asked the authors to precisely define and quantify the different types of bodies they observe. In response to this request, the authors simplified the naming of the different structures. Still, I have difficulties with the qualitative distinction that is made between the toroidal structures vs the “punctates” (actually, although I am not a native English speaker, I am not sure that using the word “punctate” as a noun is appropriate: I think it is mainly used as an adjective). These two types of structures are very difficult to distinguish on the Figure images, which makes this distinction difficult to understand and evaluate. Wouldn’t it be better (and more meaningful?) to categorize the different types of structures by their apparent size on 2D images (or volume when 3D information is available) rather than by their qualitative shape (toroidal vs punctate)? Furthermore, the authors could then use this metrics all along their manuscript to more quantitatively describe the transitions that they observe during the cell cycle and after heavy metal treatment. I believe this would be more rigorous and informative than the current classification.

2) Reviewer 1 asked the authors to “extensively rewrite” the manuscript and stressed that “the reading is sometimes difficult”. Similarly, I asked the authors to “better explain the significance of their findings”. Although some text changes have been made, the overall manuscript remains difficult to read and the conveyed messages are not always clear. In my opinion, more efforts could and should be made to more clearly explain the context of the study, the rationale of the experiments, the main findings and their biological implications. Actually, in some cases the implemented text changes did not improve clarity. For instance, I do not understand the meaning of the new sentence “the SUMO-SIM interaction, which is implicated in liquid-liquid phase separation [32], occurs only when PML-VI is SUMOylated” (page 4) which the authors added in response to a question of reviewer 1. How can there be a “SUMO-SIM” interaction if PML-VI has no SIM ? Do the authors refer to interactions with the SIM of other proteins? As another example, the authors replaced in the introduction the term “life-time” by “residence-time” (page 2). Indeed, as noted by reviewer 1, the term “life-time” was not correct. But the authors omitted to rephrase the rest of the sentence, which in my opinion is still unclear. To clarify this sentence, the authors simply need to explicitly write that what they describe is the “residence-time of PML in nuclear bodies”. Thus, I really encourage the authors to make further efforts to improve the clarity of their manuscript.

7. PLOS authors have the option to publish the peer review history of their article (what does this mean?). If published, this will include your full peer review and any attached files.

Reviewer #1: **Yes: **Olivier Coux

Reviewer #2: No

---

## [Editor Report · Decision Letter 2]

10 May 2022

Effects of arsenic on the topology and solubility of promyelocytic leukemia (PML)-nuclear bodies

PONE-D-21-36000R2

Dear Dr. Hirano,

We’re pleased to inform you that your manuscript has been judged scientifically suitable for publication and will be formally accepted for publication once it meets all outstanding technical requirements.

Kind regards,

Claude Prigent

Academic Editor

PLOS ONE
---

## [Editor Report · Acceptance letter]

13 May 2022

PONE-D-21-36000R2 

Effects of arsenic on the topology and solubility of promyelocytic leukemia (PML)-nuclear bodies 

Dear Dr. Hirano:

I'm pleased to inform you that your manuscript has been deemed suitable for publication in PLOS ONE. Congratulations! Your manuscript is now with our production department. 

Kind regards, 

on behalf of

Dr. Claude Prigent 

Academic Editor

PLOS ONE